# Self-Reported Cannabis Use and HIV Viral Control among Patients with HIV Engaged in Care: Results from a National Cohort Study

**DOI:** 10.3390/ijerph19095649

**Published:** 2022-05-06

**Authors:** Anees Bahji, Yu Li, Rachel Vickers-Smith, Stephen Crystal, Robert D. Kerns, Kirsha S. Gordon, Alexandria Macmadu, Melissa Skanderson, Kaku So-Armah, Minhee L. Sung, Fiona Bhondoekhan, Brandon D. L. Marshall, E. Jennifer Edelman

**Affiliations:** 1Department of Psychiatry, University of Calgary, Calgary, AB T2N 4N1, Canada; anees.bahji1@ucalgary.ca; 2Department of Community Health Sciences, University of Calgary, Calgary, AB T2N 4N1, Canada; 3British Columbia Centre on Substance Use, Vancouver, BC V6Z 2A9, Canada; 4Research in Addiction Medicine Scholars Program, Boston University Medical Center, Boston, MA 02118, USA; 5Hotchkiss Brain Institute, University of Calgary, Calgary, AB T2N 4N1, Canada; 6Department of Epidemiology, Brown University School of Public Health, Providence, RI 02912, USA; yu_li1@brown.edu (Y.L.); alexandria_macmadu@brown.edu (A.M.); fiona_bhondoekhan@brown.edu (F.B.); brandon_marshall@brown.edu (B.D.L.M.); 7Department of Epidemiology, University of Kentucky College of Public Health, Lexington, KY 40536, USA; rachel.vickers@uky.edu; 8Center for Health Services Research, Institute for Health, Rutgers University, Rutgers, NJ 08901, USA; scrystal@ifh.rutgers.edu; 9Department of Psychiatry, Yale School of Medicine, New Haven, CT 06511, USA; robert.kerns@yale.edu; 10VA Connecticut Healthcare System, West Haven, CT 06516, USA; kirsha.gordon2@va.gov (K.S.G.); melissa.skanderson@va.gov (M.S.); minhee.sung@yale.edu (M.L.S.); 11Department of Internal Medicine, Yale School of Medicine, New Haven, CT 06510, USA; 12Clinical Addiction Research & Education (CARE) Unit, Boston University School of Medicine, Boston, MA 02118, USA; kaku@bu.edu; 13Center for Interdisciplinary Research on AIDS, Yale School of Public Health, New Haven, CT 06511, USA

**Keywords:** cannabis, HIV, viral load, adherence

## Abstract

**Background:** The association between cannabis use and HIV-1 RNA (viral load) among people with HIV (PWH) engaged in care is unclear. **Methods:** We used data collected from 2002 to 2018 on PWH receiving antiretroviral therapy (ART) enrolled in the Veterans Aging Cohort Study. Generalized estimating equations were used to estimate associations between self-reported past-year cannabis use and detectable viral load (≥500 copies/mL), with and without adjustment for demographics, other substance use, and adherence. **Results:** Among 2515 participants, 97% were male, 66% were Black, the mean age was 50 years, and 33% had detectable HIV viral load at the first study visit. In unadjusted analyses, PWH with any past-year cannabis use had 21% higher odds of a detectable viral load than those with no past-year use (OR = 1.21; 95% CI, 1.07–1.37). However, there was no significant association between cannabis use and viral load after adjustment. **Conclusions:** Among PWH engaged in care and receiving ART, cannabis use is associated with decreased adherence in unadjusted analyses but does not appear to directly impact viral control. Future studies are needed to understand other potential risks and benefits of cannabis use among PWH.

## 1. Introduction

Cannabis use is highly prevalent among people with HIV (PWH), with approximately 77% and 34% of PWH reporting lifetime and past-year cannabis use, respectively [1]. Reasons for cannabis use among PWH include recreational and symptom management purposes, for symptoms including pain, nausea, poor appetite, depression, and anxiety [2,3].

Preclinical HIV studies have found that cannabinoids can reduce inflammation and viral load [4], while others have shown immunosuppression and elevated viral replication rates. In clinical studies, cannabis use has not been linked to mortality [5] and may be associated with higher CD4 counts [6,7] and retention in HIV care [8]. However, the relationship between cannabis use and other HIV-related outcomes (e.g., CD4 count, antiretroviral therapy (ART) adherence, and viral load) has been inconsistent and inconclusive [9]. For example, while six studies found a significant association between cannabis use and ART adherence or higher viral loads among PWH [10,11,12,13,14,15], nine found negative effects [16,17,18,19,20,21,22,23,24].

Variations in the relationship between cannabis and HIV-related outcomes may be related to differences in study populations, cannabis use measures, and confounding variables [14]. Other issues include overreliance on small samples, assessed at a single time point and a single site, limited geographic diversity, potential confounding from the method of HIV acquisition (e.g., from injection drug use versus other routes), and varying engagement in HIV care [3,5,11,14,16,25].

A key challenge is the classification of ART adherence in the cannabis–HIV relationship across extant studies. Prior work has viewed adherence as a mediator between cannabis and HIV viral load, while others have considered it a confounding variable of the cannabis–HIV relationship [26,27]. To that end, cannabis may influence HIV viral load via adherence (i.e., behavioral modification) but can also directly attenuate HIV infection progression, as demonstrated in animal models [28].

To address these knowledge deficits, this study examined the association between longitudinal measures of cannabis use and HIV viral load among PWH engaged in care in the Veterans Aging Cohort Study (VACS), a multi-site prospective cohort [29]. We hypothesized that cannabis use would be associated with a higher likelihood of having a detectable HIV viral load.

## 2. Methods

### 2.1. Study Design and Data Sources

The methods and design of the VACS survey study have been described previously [29,30]. Briefly, VACS is a multi-center, prospective cohort study of US-based PWH and controls (people without HIV), engaged in care across eight Veterans Health Administration (VA) sites in major cities in New York, Pennsylvania, Texas, Maryland, Washington, D.C., California, and Georgia. Data were obtained via annual, self-administered questionnaires spanning seven waves of data collection; survey data were then linked to VA electronic medical records, pharmacy, and laboratory data. Surveys include participant sociodemographic details, general health status, HIV outcomes, and substance use patterns. The institutional review boards at Yale University, VA Connecticut Healthcare System, and each participating site approved the study.

### 2.2. Study Population

For this analysis, we restricted our analytic sample to participants diagnosed with HIV who were receiving ART at baseline (enrolled 2002–2012 with follow-up through 2018) and who had completed a baseline survey and at least one follow-up survey. We excluded VACS participants for whom data on the outcome of interest (i.e., HIV viral load) and key covariates (i.e., CD4 cell count, VACS 2.0 Index (a validated measure of morbidity and mortality risk [31]), and ART adherence data) were unavailable. We considered participants lost to follow-up for reasons other than death if their final VACS survey occurred earlier than 2016.

### 2.3. Study Measures

***Exposure.*** The use of cannabis and other drugs was evaluated by asking how often participants had used marijuana or hashish, cocaine or crack, and other stimulants in the past year. Response options included: have never tried; no use in the last year; less than once a month; one to three times a month; one to three times a week; four to six times a week; and every day. The primary exposure variable was dichotomized to any past-year cannabis use (yes/no) assessed at baseline and in all follow-up surveys through 2018, based on the sample size and preliminary results that showed no dose–response relationship using a four-category cannabis variable.

*Outcome.* The primary outcome was dichotomized as detectable HIV viral load (≥500 vs. <500 copies/mL) based on laboratory data assessed closest to the survey date across sites and the study period [32].

Covariates. We selected additional covariates as per prior studies of PWH [3,5,11,16,17,25,33,34], including sociodemographic characteristics, general health status, prescription medications, and other substance use. We also controlled for the VACS study site and year of entry. Sociodemographic characteristics included age, sex, race, ethnicity, highest educational attainment, marital status, housing instability, urbanicity of residency [35], annual household income, and calculated Social Isolation Score (SIS). The SIS is a validated measure of social isolation for PWH that typically ranges from 0 to 8 in 0.5 increments; SIS ≥ 4 is associated with a 25% increased risk of hospitalization and a 28% increased risk of all-cause mortality [36]. HIV-related factors were CD4 cell count (cells/mm^3^) and VACS Index 2.0 score [31,37]. Scores on the VACS Index 2.0 range from 0 to 160, with each 5-point increment conferring a 30% increased risk of all-cause mortality [31].

We defined ART adherence as per prior VACS analyses: using VA pharmacy data, we calculated the duration of time the patient had ART medication available relative to the total number of days between refills for all antiretrovirals in the past year, generating a continuous measure of the percentage of days with ART medication fill, which was then dichotomized at >80% versus ≤80% [38,39,40].

General health variables included hepatitis C (HCV) co-infection, cancer, anxiety, depressive symptoms, and pain interference. HCV infection was defined based on a combination of HCV antibody status, RNA, and diagnostic codes. Cancer was defined as a history of any non-melanoma cancer as per the national VA Cancer Registry [41]. Anxiety was assessed by the HIV Symptoms Index item [42,43], asking if respondents felt anxious or nervous during the previous four weeks and the degree of impairment from such feelings. We dichotomized responses as absent (“I do not have this symptom” or “I have this symptom and it doesn’t bother me”) or present (“It bothers me a little,” “It bothers me,” or “It bothers me a lot”).

Depression was defined using the Patient Health Questionnaire-9 (PHQ-9) score of ≥10 [44,45]. Pain interference was assessed with one item from the twelve-item short-form self-report scale (SF-12) of health-related quality of life that asked: “During the past four weeks, how much did pain interfere with your normal work (including both work outside the home and housework)?” Again, response options were dichotomized (not at all or a little bit vs. moderately, quite a bit, or extremely) [46].

Other substance use categories included current cigarette smoking, unhealthy alcohol use (assessed with the Alcohol Use Disorders Identification Test-Consumption (AUDIT-C) questionnaire score of ≥3 for women and ≥4 for men [47]), stimulant use (any self-reported psychostimulant use in the past year), and self-report or prescription opioid use.

Outpatient VA pharmacy data informed prescribed medications in the year before the baseline survey date as per our prior methods [48,49]. Prescription opioid receipt was defined per morphine daily equivalents (low dose: <50 mg, high dose: ≥50 mg) and duration (short-term: <90 days supplied, long-term: ≥90 days supplied) [50]. We also examined receipt of benzodiazepines, gabapentin, and antidepressants, given that they are commonly prescribed for conditions associated with cannabinoid use [21,51,52,53,54,55,56].

### 2.4. Statistical Analyses

All statistical analyses were conducted in SAS (version 9.4), and *p*-values < 0.05 were considered statistically significant. First, we conducted descriptive statistics for participant sociodemographic characteristics across self-reported use of cannabis. Second, we examined bivariate associations between cannabis use and binary/categorical and continuous baseline characteristics with Pearson χ^2^ and Kruskal–Wallis tests, respectively, for non-parametric variables and ANOVA for normally distributed variables. Third, to determine if cannabis use was independently associated with dichotomized detectable viral load (≥500 vs. <500 copies/mL), we utilized generalized estimating equations (GEE) to account for repeated measures with a binomial distribution, generating odds ratios (ORs) with their respective 95% confidence intervals (95% CI). We ran unadjusted models and generated a series of models with stepwise covariate adjustment to assess the overall association with a detectable virus. We built a series of nested models with stepwise covariate adjustment to account for potential confounding from our covariates. In the first model, we adjusted for age, race/ethnicity, and sex. In the second model, we adjusted for unhealthy alcohol use, past-year stimulant/cocaine use, and prescribed opioid receipt. The third model added adjustment for marital status, homelessness, annual income, SIS, HCV co-infection, anxiety, depression, smoking, and antidepressant prescript. In the fourth model, we added adjustment for ART adherence. Covariates selected for inclusion in the GEE models were informed by previous cohort studies examining substance use and HIV outcomes among PWH (including prior VACS studies), and covariates significantly associated with cannabis exposure in bivariate analyses [32,37,43,49]. Time-dependent covariates considered in the GEE models were cannabis use, age, depression, unhealthy alcohol use, past-year stimulant use, opioid use, and days of antidepressant use. Baseline covariates considered in the GEE models were race, marital status, education, income, SIS, HCV co-infection, anxiety, and tobacco use. We also calculated variance inflation factors to assess multicollinearity, using a cut-off of <6 [57]. Finally, in a sensitivity analysis, we repeated the same stepwise adjustments using a three-category cannabis variable (no lifetime use, lifetime use but none in past year, past-year use).

## 3. Results

### 3.1. Eligible Participants

Among PWH in the VACS receiving ART (*n* = 2855), we excluded 340 (11.9%) participants with missing HIV viral load, VACS score, and/or self-reported items of cannabis use during the study period, yielding a final analytic sample of 2515 participants.

### 3.2. Baseline Participant Characteristics

Over the 16-year study, the mean years of follow-up were 7 (SD = 3.7), and 35% of the participants died (Table 1). The study sample consisted of mostly men (97%) with a mean age of 50 years (SD = 9 years), 65% of the participants were Black, 60% had completed some college, 40% were married or living with a partner, 40% had experienced housing instability, 95% lived in urban settings, and 50% had an annual household income less than $11,999. At the first study visit, the median CD4 cell count was 374 cells/mm^3^ (interquartile range (IQR) = 229, 569), and 33% had a detectable HIV viral load based on the baseline visit. The median VACS Index 2.0 score was 56 (IQR = 46, 66). Regarding health status, 37% had evidence of HCV co-infection, 21% received a cancer diagnosis, 37% reported anxiety, 22% reported moderate or severe depressive symptoms, and 33% endorsed pain interference. The proportions of participants with psychoactive medication receipt were as follows: 28% opioids; 15% benzodiazepines; 11% gabapentin; and 38% antidepressants. For other substance use, 76% reported smoking cigarettes, 35% screened positive for unhealthy alcohol use, and 22% reported stimulant use in the past year. Overall, 62% were adherent to ART in the sample.

### 3.3. Factors Associated with Cannabis Use: Bivariable Analyses

Cannabis use was prevalent in our sample, with 27.2% reporting any past-year use (Table 1). These frequencies did not vary substantially across subsequent VACS survey waves (Figure 1). In bivariable analyses, age, housing instability, annual income, SIS, and HIV viral load varied between those with and without past-year cannabis use (global *p*-values < 0.05, Table 1). In addition, anxiety, depression, pain interference, opioid prescription medication receipt, and substance use varied across groups (global *p*-values < 0.05). For example, depressive symptoms were only noted by 19% of PWH with no past-year cannabis use but by 28% with any past-year use (*p* < 0.0001).

### 3.4. Association between Cannabis Use and HIV Viral Load

*Unadjusted analyses for overall association with detectable viral load.* In unadjusted analyses, PWH with any past-year cannabis use had 21% greater odds of a detectable virus than those with no past-year use (OR = 1.21; 95% CI, 1.07–1.37) (Table 2, Figure 2). 

*Covariate-adjusted analyses for an independent association between cannabis use and detectable HIV viral load.* In a series of models, sequentially adjusting for potential confounders, past-year cannabis use was not associated with the presence of a detectable HIV viral load (Table 2, Figure 2).

*Sensitivity analyses*. In sensitivity analyses using the three-level cannabis exposure variable (no lifetime use, no past-year use, any past-year use), any past-year use was associated with a detectable virus (OR = 1.18; 95% CI, 1.01–1.36), while reported lifetime use but no past-year use (OR = 0.95; 95% CI, 0.84–1.08) was not associated with the detectable virus compared to no lifetime cannabis use (Appendix A). However, the combined variable of cannabis use at any time was not associated with a detectable HIV viral load (Appendix A).

## 4. Discussion

To our knowledge, this is one of the largest multi-site cohort studies to examine longitudinal patterns of cannabis use patterns and their association with HIV viral load among PWH receiving ART. Almost a third of the sample used cannabis at the first study visit, and overall use remained stable over the 16-year study period. Unadjusted models identified a 21% increase in the odds of having a detectable viral load among participants who reported any past-year cannabis use compared to those with no past-year cannabis use. However, the association was no longer statistically significant after adjusting for sociodemographics, substance use, clinical characteristics, and ART adherence. 

Numerous studies spanning diverse settings have shown that PWH who use drugs have lower rates of ART initiation [58], higher viral loads [59], and greater mortality [60]. However, it is interesting that these relationships do not appear to consistently apply to cannabis use [61]. In this context, and as cannabis use is highly prevalent among PWH [2,3], the finding that cannabis use was not reliably associated with HIV viral load—particularly after accounting for other covariates, including ART adherence—is reassuring. Furthermore, our findings are consistent with several previous studies of PWH, such as one by Lake et al., which found that daily cannabis use was not associated with HIV viral load among people who use drugs (PWUD), except when combined with heavy episodic drinking [14]. In addition, Lorenz et al. found that cannabis use did not predict poor ART adherence or virologic suppression [15].

Similarly, Slawson et al. found that high-intensity cannabis use was not associated with adherence to ART [10]. However, it is important to emphasize that most of these studies, including the present analyses, have focused on cannabis use and not cannabis use disorder (CUD), cannabis abuse (CA), or cannabis dependence (CD). In addition, only one prior study has examined the association between CD and HIV-related outcomes, finding that CD, and not cannabis use alone, was associated with poorer ART adherence among PWH [16]. Ultimately, these findings highlight the importance of distinguishing cannabis use from CUD, CA, and CD. 

Our analyses considered ART adherence as a potential confounder of the relationship between cannabis use and HIV viral load. However, it is also possible that ART adherence is a mediator of the cannabis use and HIV viral load relationship. Although this was not the approach taken by the present study, prior work has found no association between cannabis and adherence to ART [10]. There may be other indirect effects of cannabis use on HIV viral load, such as behavioral effects of cannabis on adherence that, in turn, could be associated with viral load [16]. In addition, given that cannabis use has been reported by some PWH for sexual enhancement [62,63], it is theoretically possible that cannabis may drive unprotected sex between PWH, which may somehow impact the diversity of viral exposure and possibly, viral replication. Furthermore, as cannabis use varies by alcohol use, an established driver of poor ART adherence [64], alcohol may be a major potential driver of a detectable viral load among PWH who reported past-year cannabis use.

In addition to the main findings on cannabis use and HIV viral load, several important and statistically significant associations may have driven our unadjusted findings. For example, we found a higher prevalence of other categories of substance use (stimulants, alcohol, cigarettes, prescribed opioids) and a higher prevalence of anxiety, depression, and pain among those with past-year cannabis use. As cannabis use is associated with using other psychoactive substances and may be used to self-medicate for various health conditions, these associations may have driven the unadjusted findings to a certain extent. Similarly, it is possible that the co-use of other psychoactive substances, such as alcohol, tobacco, and stimulants, may be relevant in individuals with past-year cannabis use and may also be partly responsible for the unadjusted associations. For example, in Table 2, after covariate adjustment, there was a strong association between past-year stimulant use and a detectable viral load.

Ultimately, these findings should also be contextualized within a growing body of preclinical data that has identified a potential role of cannabinoids through the human endocannabinoids system on HIV disease pathophysiology. For example, some animal studies have shown that exposure to tetrahydrocannabinol (THC)—the active component of whole cannabis—appears to be associated with lower viral loads in HIV-infected rhesus macaques [65]. Furthermore, some experimental studies have identified a direct antiviral effect of cannabinoids, which appears to mediate inhibition of HIV-1 expression in microglial cells, the primary immune cell infected in the human nervous system [66,67]. 

### 4.1. Limitations

Our study is also subject to limitations. First, as VACS is a primarily male, non-random sample, our findings may not generalize to PWH in other settings, particularly those not engaged in VA care or receiving ART. Second, while we used objective measures wherever possible, including our measures of ART adherence and HIV viral load, we relied on self-report to obtain the main exposure variable of interest (i.e., cannabis use). While misclassification from underreporting due to recall or response biases may have occurred, it is unlikely that self-report of cannabis use would affect the association with viral load, given that self-report measures of drug use among PWUD are generally reliable and valid [68,69,70]. Third, we did not capture cannabis dosage, potency, consumption modalities, cannabis use disorder, or cannabis legalization status across states. Although not a specific limitation of the present study, it raises the importance of having a standard unit dose for cannabis research [71]. Fourth, while GEE models can help analyze repeated measures over time, there is a risk of unmeasured confounding.

### 4.2. Future Research

The current findings affirm that cannabis use does not appear to be associated with HIV viral load among PWH engaged in care after adjustment. Nonetheless, ongoing investigation of other potential risks (e.g., cardiovascular disease, lung disease, mood and anxiety disorders) and benefits (e.g., pain control) associated with cannabis are needed to inform clinical practice. In addition, future research may consider whether cannabis use moderates opioid and other substance use among PWH engaged in care [32], explore the association between cannabis use and earlier stages of the HIV care continuum (e.g., ART receipt), and consider more nuanced assessments of cannabis use, including dosage, modality, type, indication, and context (e.g., CUD, CD, or CA versus use alone).

## 5. Conclusions

There was no association between cannabis use and HIV viral load after adjusting for sociodemographic characteristics, other substance use, and adherence. Unmeasured variables may confound the observed relationships. Possible indirect mediating effects of cannabis use on viral load may exist, such as effects of cannabis on adherence, that in turn could be associated with viral load and, as such, require investigation. At present, clinicians should remain cautious about the impact of cannabis use on HIV-related outcomes among PWH. 

## Figures and Tables

**Figure 1 ijerph-19-05649-f001:**
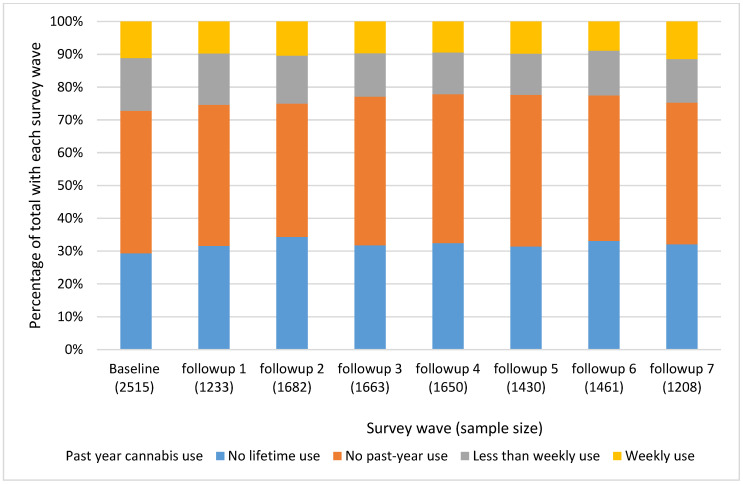
Self-reported cannabis use frequency across VACS survey waves. *Notes*: Numbers underneath the survey wave indicate the number of participants who completed each survey wave (e.g., 2515 participants were identified at the baseline assessment).

**Figure 2 ijerph-19-05649-f002:**
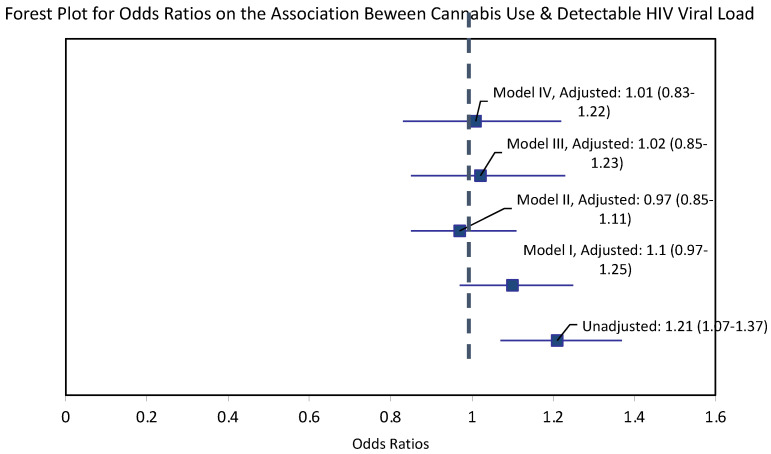
Association between cannabis use and detectable HIV viral load: Results from logistic regression analyses.

**Table 1 ijerph-19-05649-t001:** Baseline sociodemographic and clinical characteristics associated with cannabis use categories among people living with HIV receiving antiretroviral therapy in the VACS cohort (2002–2018).

Characteristic	Overall (*n* = 2515)	No Past-Year Use (*n* = 1830, 72.8%)	Past-Year Use (*n* = 685, 27.2%)	Global *p*-Value
**Demographics**				
**Age, mean (SD)**	50.1 (8.8)	50.6 (9.0)	48.7 (8.2)	**<0.0001**
**Gender, *n* (%)**				0.494
Male	2447 (97.3)	1783 (97.4)	664 (96.9)	
Female	68 (2.7)	47 (2.6)	21 (3.1)	
**Race/ethnicity, *n* (%)**				0.160
Non-Hispanic White	524 (20.8)	363 (19.8)	161 (23.5)	
Non-Hispanic Black	1646 (65.5)	1214 (66.3)	432 (63.1)	
Hispanic (any race)	247 (9.8)	185 (10.1)	62 (9.1)	
Other (multiple race or unknown)	98 (3.9)	68 (3.7)	30 (4.4)	
**Education, *n* (%)**				0.077
High school or less	987 (39.7)	736 (40.8)	251 (36.9)	
Some college or more	1500 (60.3)	1070 (59.3)	430 (63.1)	
**Marital status, *n* (%)**				0.159
Never married	610 (24.6)	440 (24.4)	170 (25.2)	
Married/living with a partner	981 (39.6)	734 (40.7)	247 (36.7)	
Divorced/widowed	885 (35.7)	628 (34.9)	257 (38.1)	
**Housing instability ever, *n* (%)**	982 (39.3)	680 (37.4)	302 (44.3)	**0.002**
**Location of residence, *n* (%)**				0.512
Urban	2368 (95.0)	1726 (95.2)	642 (94.3)	
Suburban	70 (2.8)	50 (2.8)	20 (2.9)	
Rural	56 (2.3)	37 (2.0)	19 (2.8)	
**Annual income, *n* (%)**				**0.013**
<$11,999	1213 (49.8)	859 (48.6)	354 (53.2)	
$12,000–$49,999	1038 (42.6)	761 (43.0)	277 (41.6)	
≥$50,000	184 (7.6)	149 (8.4)	35 (5.3)	
**Social Isolation Score**				**0.032**
<4	680 (27.0)	516 (28.2)	164 (23.9)	
≥4	1835 (73.0)	1314 (71.8)	521 (76.1)	
**HIV-related factors**				
CD4 cell count, cells/mm^3^, median (IQR)	374 (228, 568)	377 (231, 578)	365 (219, 549)	0.295
HIV viral load <500 copies/mL, *n* (%)	1690 (67.2)	1259 (68.8)	431 (62.9)	**0.005**
VACS Index 2.0 score, median (IQR)	56 (46, 66)	56 (46, 66)	56 (46, 67)	0.109
ART adherent, *n* (%)	1551 (61.7)	1164 (63.6)	387 (56.5)	**0.001**
**Other health conditions and status, *n* (%)**				
HCV co-infection	929 (36.9)	664 (36.3)	265 (38.7)	0.267
Any cancer	524 (20.8)	375 (20.5)	149 (21.8)	0.489
Anxiety symptoms	905 (37.1)	619 (34.9)	286 (42.9)	**0.0003**
Depressive symptoms	534 (21.5)	347 (19.2)	187 (27.5)	**<0.0001**
Pain interference	830 (33.3)	582 (32.1)	248 (36.4)	**0.043**
**Other substance use, *n* (%)**				
Smokes cigarettes	1915 (76.1)	1337 (73.1)	578 (84.4)	**<0.0001**
Unhealthy alcohol use	876 (34.8)	599 (32.7)	277 (40.4)	**0.0003**
Past-year stimulants or cocaine	538 (21.4)	252 (13.8)	286 (41.8)	**<0.0001**
**Prescribed opioid receipt, *n* (%)**				**0.001**
No opioid receipt	1802 (71.7)	1346 (73.6)	456 (66.6)	
Short-term + low dose	449 (17.9)	308 (16.8)	141 (20.6)	
Short-term + high dose	40 (1.6)	33 (1.8)	7 (1.0)	
Long-term + low dose	159 (6.3)	100 (5.5)	59 (8.6)	
Long-term + high dose	65 (2.6)	43 (2.4)	22 (3.2)	
**Prescribed benzodiazepine**				0.336
None	2147 (85.4)	1573 (86.0)	574 (83.8)	
Low dose	269 (10.7)	190 (10.4)	79 (11.5)	
High dose	99 (3.9)	67 (3.7)	32 (4.7)	
**Prescribed gabapentin**				0.320
None	2240 (89.1)	1639 (89.6)	601 (87.7)	
Low dose	127 (5.1)	91 (5.0)	36 (5.3)	
High dose	148 (5.9)	100 (5.5)	48 (7.0)	
**Prescribed antidepressant**				**0.017**
None	1557 (61.9)	1161 (63.4)	396 (57.8)	
Short-term	314 (12.5)	227 (12.4)	87 (12.7)	
Long-term	644 (25.6)	442 (24.2)	202 (29.5)	
**Site**				0.193
Atlanta	403 (16.0)	299 (16.3)	104 (15.2)	
Bronx	258 (10.3)	190 (10.4)	68 (9.9)	
Houston	335 (13.3)	242 (13.2)	93 (13.6)	
Los Angeles	334 (13.3)	225 (12.3)	109 (15.9)	
New York	399 (15.9)	295 (16.1)	104 (15.2)	
Baltimore	282 (11.2)	214 (11.7)	68 (9.9)	
Washington DC	408 (16.2)	302 (16.5)	106 (15.5)	
Pittsburgh	96 (3.8)	63 (3.4)	33 (4.8)	
**Calendar year**				0.084
2002–2006	781 (31.5)	583 (32.3)	198 (29.3)	
2007–2011	976 (39.3)	717 (39.7)	259 (38.3)	
2012–2017	724 (29.2)	505 (28.0)	219 (32.4)	
Average follow-up years, mean (SD)	7.0 (3.7)	6.9 (3.7)	7.3 (3.7)	**0.030**
Died during study	904 (35.9)	666 (36.4)	238 (34.7)	0.443

**Table 2 ijerph-19-05649-t002:** Generalized estimating equation (GEE) analysis for the association between frequency of cannabis use with detectable HIV viral load status (<500 copies/mL vs. ≥500 copies/mL) among *n* = 2515 PWH engaged in care. Results are presented as odds ratios (OR) from the unadjusted analysis, adjusted for all covariates, and adjusted for all covariates plus adherence to antiretroviral therapy.

	Unadjusted Odds Ratios [95% CI]	Model I, Adjusted Odds Ratios [95% CI]	Model II, Adjusted Odds Ratios [95% CI]	Model III, Adjusted Odds Ratios [95% CI]	Model IV, Adjusted Odds Ratios [95% CI]
**Past-year cannabis**	**1.21 (1.07–1.37)**	1.10 (0.97–1.25)	0.97 (0.85–1.11)	1.02 (0.85–1.23)	1.01 (0.83–1.22)
**Age**		**0.95 (0.94, 0.95)**	**0.95 (0.94–0.95)**	**0.96 (0.95–0.97)**	**0.96 (0.95–0.97)**
**Race/ethnicity**					
Non-Hispanic White		Ref	Ref	Ref	Ref
Non-Hispanic Black		**1.21 (1.02–1.43)**	1.15 (0.97–1.37)	1.21 (0.97–1.50)	1.12 (0.90–1.40)
Hispanic (any race)		0.85 (0.65–1.10)	0.82 (0.64–1.07)	0.93 (0.67–1.29)	0.93 (0.67–1.30)
Other (multiple race or unknown)		1.00 (0.69–1.44)	0.96 (0.66–1.40)	1.19 (0.76–1.86)	1.14 (0.72–1.79)
**Sex**					
Male		Ref	Ref	Ref	Ref
Female		**0.62 (0.40–0.94)**	**0.62 (0.41–0.95)**	0.64 (0.36–1.14)	**0.56 (0.32–0.97)**
**Unhealthy alcohol use**					
Yes			**1.14 (1.01–1.29)**	**0.83 (0.70–0.98)**	**0.78 (0.66–0.93)**
No			Ref	Ref	Ref
**Past-year stimulant/cocaine use**					
Yes			**1.56 (1.35–1.79)**	**1.45 (1.18–1.79)**	**1.37 (1.11–1.69)**
No			Ref	Ref	Ref
**Opioid Use**					
No opioid receipt			Ref	Ref	Ref
Short-term + low dose			1.02 (0.90–1.15)	0.98 (0.81–1.20)	0.97 (0.80–1.19)
Short-term + high dose			1.04 (0.67–1.60)	1.86 (0.99–3.49)	1.72 (0.92–3.24)
Long-term + low dose			1.21 (0.99–1.49)	1.27 (0.93–1.73)	1.25 (0.90–1.72)
Long-term + high dose			0.97 (0.70–1.35)	0.82 (0.50–1.35)	0.85 (0.52–1.40)
**Marital Status**					
Never married				Ref	Ref
Married/living with a partner				1.12 (0.88–1.43)	1.10 (0.86–1.42)
Divorced/widowed				1.03 (0.80–1.33)	1.01 (0.78–1.30)
**Homeless**					
Yes				1.01 (0.84–1.22)	0.98 (0.81–1.18)
No				Ref	Ref
**Income**					
<$11,999				Ref	Ref
$12,000–$49,999				1.08 (0.90–1.30)	1.07 (0.89–1.29)
≥$50,000				1.04 (0.74–1.47)	1.05 (0.75–1.49)
**Social score**					
<4				Ref	Ref
≥4				1.01 (0.81–1.26)	1.03 (0.82–1.29)
**HCV**					
Yes				0.98 (0.82–1.17)	0.93 (0.78–1.12)
No				Ref	Ref
**Anxiety**					
Yes				1.04 (0.86–1.27)	1.02 (0.84–1.24)
No				Ref	Ref
**Depression**					
Yes				1.13 (0.92–1.40)	1.07 (0.86–1.32)
No				Ref	Ref
**Smoking**					
Yes				**0.81 (0.66–0.99)**	0.83 (0.67–1.01)
No				Ref	Ref
**Antidepressant**					
None				Ref	Ref
Short-term				1.24 (0.97–1.57)	1.23 (0.96–1.58)
Long-term				1.13 (0.92–1.38)	**1.27 (1.03–1.55)**
**ART adherence**					
Yes					**0.41 (0.35–0.48)**
No					Ref

Bold indicates statistically significant results. Time-dependent covariates considered in GEE models were cannabis use, age, depression, unhealthy alcohol use, past-year stimulant use, opioid use, and days of antidepressant use. Baseline covariates considered in GEE models were race, marital status, education, income, SIS, HCV co-infection, anxiety, and tobacco use.

## Data Availability

Data requests should be submitted by email to the corresponding author; data will be available pending VACS approval.

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
