# Peer review of "Self-Reported Cannabis Use and HIV Viral Control among Patients with HIV Engaged in Care: Results from a National Cohort Study"

_ijerph, 2022, doi:10.3390/ijerph19095649_

Round 1

Reviewer 1 Report

In this study the authors examined the association between longitudinal measures of cannabis use and HIV viral load among PWH engaged in care in the Veterans Aging Cohort Study. The manuscript is well-written, and the tables are adequately presented. The analysis was done using appropriate statistical method. However, there are some concerns that I highly recommend to address:

  1. My first concern refers to the lack of information on the amount of cannabis consumed by the participants. What is considered minimum or maximum doses?
  2. I suggest adjusting the tittle to increase adequation with the showing results.
  3. I suggest rewriting the introduction in link with the results shown.

Author Response

Comment #1. My first concern refers to the lack of information on the amount of cannabis consumed by the participants. For example, what is considered minimum or maximum doses?

Response #1. In this study, the participants’ cannabis consumption was quantified by frequency rather than dose, as described in the Methods. Ultimately, this is a general limitation of cannabis research, emphasizing the need for a standardized “dose” or “unit” of cannabis consumption. Therefore, we noted this as a limitation and modified this to now read:

Page 11, paragraph 2: Third, we did not capture cannabis dosage, potency, consumption modalities, cannabis use disorder, or cannabis legalization status across states.

Comment #2. I suggest adjusting the title to show results.

Response #2. While we appreciate this suggestion and have considered it, we prefer to keep the original title, “Self-reported cannabis use and HIV viral control among patients with HIV engaged in care: Results from a national cohort study,”  given our findings of a statistically significant association in unadjusted, but not adjusted analyses.

Comment #3. In addition, I suggest rewriting the introduction in the link with the results shown.

Response #3. Thank you for this suggestion; we believe the current introduction provides relevant background and rationale for the research question addressed in this manuscript. We would be happy to revise upon more specific guidance and at the Editor’s discretion. 

Reviewer 2 Report

In this manuscript “Self-Reported Cannabis Use and HIV Viral Control among Patients with HIV Engaged in Care: Results from a National Cohort Study” by Bahji et al. authors used a large cohort of more than 2,500 PLWH followed for 16 years. This is a very interesting work that shows a lack of association between cannabis consumption and HIV load control. The manuscript is very well written, and I appreciated the limitations paragraph at the end. I have one comment I would like authors to respond.

  1. Did you attempt to do a first analysis without any categorization and adjustments? Pure correlative analysis between consumption and viral loads?
  2. Have you performed the analysis only including those who had undetectable baseline HIV load? It is possible this may be a confounding factor that is carried over the analysis.

Author Response

Comment #1. Did you attempt to do the first analysis without any categorization and adjustments? Pure correlative analysis between consumption and viral loads?

Response #1. Yes, the first analysis was a crude (unadjusted) logistic regression to assess the overall association between cannabis use and the presence of a detectable viral load. This is what is captured in the “unadjusted” models and results. In addition, we have added a figure to demonstrate these associations (see Reviewer #3, Comment #1).

Comment #2. Have you performed the analysis only including those who had undetectable baseline HIV load? This may be a confounding factor that is carried over the analysis.

Response #2. Our goal was to assess the association between cannabis use on HIV viral load among those engaged in care and receiving ART. Therefore, we restricted analyses to those with ART at baseline regardless of HIV viral load. In response to this question, we examined the baseline characteristics of those with an undetectable vs. detectable HIV viral load at baseline; we believe it would be overly restrictive to limit analyses to those with baseline undetectable viral load. However, we have provided an additional table (below) that compares the characteristics by baseline HIV viral load (<500 copies/mL vs. (≥500 copies/mL) status. We are happy to add these findings to the paper at the Editor’s discretion.

In addition, we have expanded our suggested directions for future research:

Page 11, paragraph 3: In addition, future research may consider whether cannabis use moderates opioid and other substance use among PWH engaged in care [32], explore the association between cannabis use and earlier stages of the HIV care continuum (e.g., ART receipt) consider more nuanced assessments of cannabis use, including dosage, modality, type, indication, and context (e.g., CUD, CD, or CA versus use alone).

Supplementary Table 3. Sociodemographic and clinical characteristics associated with cannabis use categories among people living with HIV receiving antiretroviral therapy in the VACS cohort (2002-2018) comparing those with an undetectable viral load (<500 copies/mL) to those with a detectable viral load (≥500 copies/mL).

Characteristic

Overall
(N= 2515)

<500 copies/mL (N=1690, 67.2%)

>=500 copies/mL (N=825, 32.8%)

Global
p-value

Demographics

Age, mean (SD)

50.1 (8.8)

51.1 (8.9)

48.0 (8.3)

<0.0001

Gender, n (%)

0.936

Male

2447 (97.3)

1644 (97.3)

803 (97.3)

Female

68 (2.7)

46 (2.7)

22 (2.7)

Race/ethnicity, n (%)

0.085

Non-Hispanic White

524 (20.8)

373 (22.1)

151 (18.3)

Black

1646 (65.5)

1078 (63.8)

568 (68.9)

Hispanic

247 (9.8)

171 (10.1)

76 (9.2)

Other

98 (3.9)

68 (4.0)

30 (3.6)

Education, n (%)

0.420

High school or less

987 (39.7)

672 (40.2)

315 (38.6)

Some college or more

1500 (60.3)

998 (59.8)

502 (61.4)

Marital status, n (%)

0.255

Never married

610 (24.6)

421 (25.3)

189 (23.3)

Married/living with a partner

981 (39.6)

666 (40.0)

315 (38.8)

Divorced/widowed

885 (35.7)

577 (34.7)

308 (37.9)

Housing instability ever, n (%)

982 (39.3)

632 (37.7)

350 (42.5)

0.019

Location of residence, n (%)

0.357

Urban

2368 (95.0)

1589 (95.1)

779 (94.7)

Suburban

70 (2.8)

42 (2.5)

28 (3.4)

Rural

56 (2.3)

40 (2.4)

16 (1.9)

Annual income, n (%)

0.475

<$11,999

1213 (49.8)

803 (49.2)

410 (51.0)

$12,000-$49,999

1038 (42.6)

698 (42.8)

340 (42.3)

≥$50,000

184 (7.6)

130 (8.0)

54 (6.7)

Social Isolation Score

0.150

<4

680 (27.0)

472 (27.9)

208 (25.2)

≥4

1835 (73.0)

1218 (72.1)

617 (74.8)

HIV-related factors

CD4 cell count, cells/mm3, median

(IQR)

374

(228, 568)

427.5 (276, 622)

282 (144, 446)

<0.0001

VACS Index 2.0 score, median (IQR)

56 (46, 66)

53 (44, 64)

61 (53, 72)

<0.0001

ART adherent, n (%)

1551 (61.7)

1158 (68.5)

393 (47.6)

<0.0001

Other health conditions and status, n (%)

HCV co-infection

929 (36.9)

634 (37.5)

295 (35.8)

0.391

Any cancer

524 (20.8)

354 (21.0)

170 (20.6)

0.843

Anxiety symptoms

905 (37.1)

592 (36.1)

313 (39.1)

0.150

Depressive symptoms

534 (21.5)

334 (20.0)

200 (24.4)

0.012

Pain interference

830 (33.3)

542 (32.5)

288 (35.0)

0.199

Other substance use, n (%)

Smokes cigarettes

1915 (76.1)

1305 (77.2)

610 (73.9)

0.070

Unhealthy alcohol use

876 (34.8)

604 (35.7)

272 (33.0)

0.171

Past year stimulants or cocaine

538 (21.4)

319 (18.9)

219 (26.6)

<0.0001

Prescribed opioid receipt, n (%)

0.198

No opioid receipt

1802 (71.7)

1215 (71.9)

587 (71.2)

Short-term + low dose

449 (17.9)

303 (17.9)

146 (17.7)

Short-term + high dose

40 (1.6)

25 (1.5)

15 (1.8)

Long-term + low dose

159 (6.3)

97 (5.7)

62 (7.5)

Long-term + high dose

65 (2.6)

50 (3.0)

15 (1.8)

Prescribed benzodiazepine

0.840

None

2147 (85.4)

1445 (85.5)

702 (85.1)

Low dose

269 (10.7)

177 (10.5)

92 (11.2)

High dose

99 (3.9)

68 (4.0)

31 (3.8)

Prescribed gabapentin

0.462

None

2240 (89.1)

1501 (88.8)

739 (89.6)

Low dose

127 (5.1)

83 (4.9)

44 (5.3)

High dose

148 (5.9)

106 (6.3)

42 (5.1)

Prescribed antidepressant

0.121

None

1557 (61.9)

1064 (63.0)

493 (59.8)

Short term

314 (12.5)

196 (11.6)

118 (14.3)

Long term

644 (25.6)

430 (25.4)

214 (25.9)

Site

<0.0001

Atlanta

403 (16.0)

191 (11.3)

212 (25.7)

Bronx

258 (10.3)

181 (10.7)

77 (9.3)

Houston

335 (13.3)

212 (12.5)

123 (14.9)

Los Angeles

334 (13.3)

243 (14.4)

91 (11.0)

New York

399 (15.9)

310 (18.3)

89 (10.8)

Baltimore

282 (11.2)

204 (12.1)

78 (9.5)

Washington DC

408 (16.2)

272(16.1)

136 (16.5)

Pittsburgh

96 (3.8)

77 (4.6)

19 (2.3)

Calendar year

0.089

2002-2006

781 (31.5)

519 (31.1)

262 (32.2)

2007-2011

976 (39.3)

680 (40.8)

296 (36.4)

2012-2017

724 (29.2)

469 (28.1)

255 (31.4)

Average follow-up years, mean (SD)

7.0 (3.7)

7.1 (3.6)

7.0 (3.9)

0.478

Died during study

904 (35.9)

569 (33.7)

335 (40.6)

0.001

Reviewer 3 Report

Bahji A et. al investigates the  association between cannabis use and HIV-1 RNA (viral load) among people with HIV (PWH).  The finding suggested that among people living with HIV who are currently engaged in care and receiving ART, cannabis use is associated with decreased adherence in unadjusted analyses, but does not have direct impact on the viral control.

The main plus point of this study is high sample size.

However, few minor modification is required.

I would request to either plot one graphical representation or make a separate table of table no 1 subsection (HIV related factor vs cannabis use categories).

Check for few typo errors.

Author Response

Comment #1. I would request to either plot one graphical representation or make a separate table of the table with no one subsection (HIV-related factors vs. cannabis use categories).

Response #1. Thank you for this suggestion – we agree that figures are helpful! Per this feedback, we have created forest plots to summarize the unadjusted and adjusted models (Figure 2, corresponding to Table 2; and Supplementary Figure 1, corresponding to Supplementary Table 2).

Comment #2. Check for a few typo errors.

Response #2. We have edited the manuscript extensively for typos and grammatical errors.